# LARGE LANGUAGE MODELS CAN *Share* IMAGES, TOO!

## ABSTRACT

This paper explores the *image-sharing* capability of Large Language Models (LLMs), such as InstructGPT, ChatGPT, and GPT-4, in a zero-shot setting, without the help of visual foundation models. Inspired by the two-stage process of image-sharing in human dialogues, we propose a two-stage framework that allows LLMs to predict potential image-sharing turns and generate related image descriptions using our effective restriction-based prompt template. With extensive experiments, we unlock the *image-sharing* capability of LLMs in zero-shot prompting, with GPT-4 achieving the best performance. Additionally, we uncover the emergent *image-sharing* ability in zero-shot prompting, demonstrating the effectiveness of restriction-based prompts in both stages of our framework. Based on this framework, we augment the PhotoChat dataset with images generated by Stable Diffusion at predicted turns, namely PhotoChat++. To our knowledge, this is the first study to assess the *image-sharing* ability of LLMs in a zero-shot setting without visual foundation models. The source code and the dataset will be released after publication.

## 1  INTRODUCTION

People often share a variety of images during interactions via instant messaging tools. In practice theory, this is referred to as *photo-sharing* behavior (Lobinger, 2016), which is interpreted as a communicative practice. From now on, we refer to this as *image-sharing* behavior, given that "image" is a broader concept than "photo," thereby providing more flexibility to language models. This behavior involves two or more individuals sharing images for various purposes, such as discussion or self-expression, during a dialogue. For example, while conversing about pets with a friend, one might share an image of their pet (e.g., a dog) to talk about the image itself. Hence, the capability to share images is also necessary for a multi-modal dialogue model to enhance social bonding (rapport) with interlocutors.

However, in the multi-modal dialogue domain, most previous studies have primarily focused on image-grounded dialogues, where two persons talk about given images (Antol et al., 2015; Das et al., 2017; Shuster et al., 2020), which usually happens after sharing an image. To address image-sharing behavior, recent studies have actively proposed multi-modal dialogue datasets (Lee et al., 2021; Zang et al., 2021; Feng et al., 2022) and multi-modal dialogue models (Zang et al., 2021; Koh et al., 2023) with the help of large language models (Zhang et al., 2022a; OpenAI, 2023a) and visual foundation models (Radford et al., 2021; Li et al., 2023). Contrary to these prior studies, we believe that large language models, which do not contain the ability of visual understanding, can share relevant images to some degree without any help of visual foundation models.

Large Language Models (LLMs), such as GPT-3 (Brown et al., 2020), InstructGPT (Ouyang et al., 2022), ChatGPT (OpenAI, 2023a), GPT-4 (OpenAI, 2023b), have shown surprising zero-/few-shot performance on various NLP tasks, such as dialogue (Kim et al., 2022), theory-of-mind (ToM) (Sap et al., 2022; Kosinski, 2023), and complex reasoning (Wei et al., 2022b; Kojima et al., 2022), through variant strategies of "prompt engineering." Prompt engineering has unlocked the potential of language models for various unseen tasks by skillfully manipulating input prompts with instructions. In addition to NLP tasks, recent studies have attempted to utilize the power of large language models for image classification (Yang et al., 2022; Pratt et al., 2022; Menon & Vondrick, 2022; Zhang et al., 2023) and multi-modal learning (Koh et al., 2023; Wu et al., 2023; Huang et al., 2023). This work primarily focuses on multi-modal dialogue, especially *image-sharing behavior* (Zang et al., 2021), which frequently occurs in social dialogues.

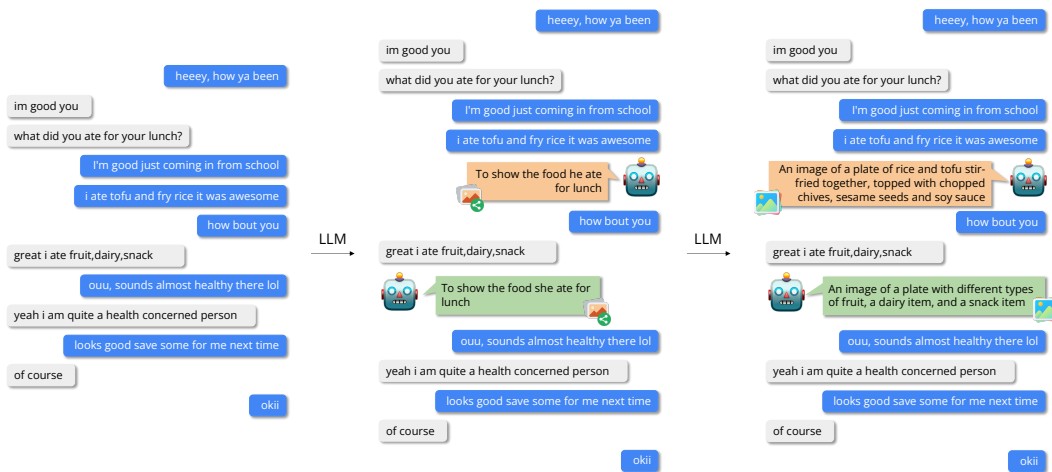

Figure 1: **An illustration of our proposed approach.** LLM predicts image-sharing turns (🤖), along with the speaker and rationale. Then, LLM generates a relevant image descriptions (🖼) for each image-sharing turn.

To enable LLMs to exhibit this *image-sharing* behavior, we ponder: *How do humans share an image during a dialogue?* We posit that our process of image-sharing internally operates through a two-stage system: (1) *when to share* and (2) *what to share*. Specifically, we first discern the appropriate moment for sharing an image by understanding the dialogue context and considering the interlocutor's mental state. Then, we share appropriate images at that moment, either by searching the internet or using photos taken on our mobile devices. Based on this system, we try to solve the existing multi-modal dialogue task as a conditional text generation task.

This work explores whether LLMs contain the *image-sharing* capability without the help of visual foundation models, primarily focusing on a zero-shot performance. To this end, we introduce a two-stage framework: (1) predicting all possible image-sharing turns and (2) generating image descriptions - to unlock this *image-sharing* capability of LLMs through in-context zero-shot learning. To elicit this *image-sharing* capability of LLM at each stage, we present a restriction-based prompt by adding a `Restrictions:` token. By leveraging our framework and restrictions-based prompt, we test the capability of LLM on PhotoChat (Zang et al., 2021) dataset, which is a multi-modal dialogue dataset constructed via crowdsourcing. In our experiments, GPT-4 (OpenAI, 2023b) achieves the best zero-shot performance in the image-sharing turn prediction task. Notably, we reveal that the image-sharing ability is an emergent ability (Wei et al., 2022a) in a zero-shot prompting. Extensive experiments demonstrate that the restriction-based prompt is effective in both stages. In addition, using our framework, we augment the existing dataset, PhotoChat++, by generating image descriptions from LLMs and corresponding images from visual foundation models. Through the generalization experiment, we observe that PhotoChat++ can enhance the generalization performance on unseen multi-modal dialogue datasets.

In summary, our main contributions are as follows: 1) We propose a two-stage framework and restriction-based prompt to evaluate the *image-sharing* ability of LLM in a zero-shot setting. 2) Experimental results show that LLMs achieve competitive zero-shot performance for both stages, even without additional training, compared to the existing method. 3) To the best of our knowledge, this is the first study to test the *image-sharing* capability of LLMs through zero-shot prompting without the aid of visual foundation models.

## 2 METHOD

In this section, we introduce a two-stage framework to unlock the *image-sharing* capabilities of LLMs in a zero-shot prompting. Our framework consists of two main stages that are designed to perform two intuitions: (1) *when to share* and (2) *what to share*.

## 2.1 PRELIMINARY: PHOTOCHAT

We select the PhotoChat (Zang et al., 2021) dataset, constructed through crowdsourcing, for assessing the *image-sharing* capability of LLMs. This dataset contains 10k multi-modal dialogues, where each dialogue $\mathcal{D} = \{(u_1, s_1), ..., (u_{t-1}, s_{t-1}), (i_t, s_t), (u_{t+1}, s_{t+1}), ..., (u_N, s_N)\}$ in the dataset contains only one image $i_t$ to be shared at turn $t$. The $N$ and $s_j \in \{0, 1\}$ denote the number of dialogue turns and speaker information, respectively. In addition, they defined two tasks by decomposing the image-sharing behavior — a photo-sharing intent prediction task and an image retrieval task. The formulations are described as follows.

**Photo-Sharing Intent Prediction.** Given the dialogue history $(u_j)_1^{t-1}$ and the corresponding speaker information $(s_j)_1^{t-1}$, this task aims to predict a turn $t$ in the binary classification formulation, where the label $y \in \{0, 1\}$.

**Image Retrieval.** Given the dialogue history $(u_j)_1^{t-1}$ and the corresponding speaker information $(s_j)_1^{t-1}$, this task aims to retrieve most appropriate image at turn $t$ from the image candidate set.

## 2.2 DO WE ONLY NEED ONE IMAGE PER DIALOGUE?

In everyday conversations, the opportunity to share an image can occur at various turns, depending on the person involved. For instance, as illustrated in Figure 1, speakers can share images at the utterance of "*i ate tofu and fry rice it was awesome*" and "*greate i ate fruit,dairy,snack*". However, the PhotoChat dataset contains only one image per dialogue, which does not fully reflect the nuances of real-life social interactions. Therefore, we aim to predict all potential image-sharing turns using LLMs and augment the PhotoChat dataset, called PhotoChat++ (in Section 4.5).

## 2.3 🚫 RESTRICTION-BASED PROMPT TEMPLATE

To elicit the *image-sharing* ability of LLM in a zero-shot setting, we manually construct a prompt template for both stages. In a pilot study, we constructed an initial prompt template based on findings from previous works. For example, we follow the multiple choice prompting (Robinson et al., 2022), simply including `Q:`, `A:` tokens, or `Options:` (Wei et al., 2021). However, these approaches did not significantly enhance performance, suggesting we could not fully exploit the LLM's capacity for *image-sharing* behavior. To unlock the *image-sharing* capability of LLMs, as shown in Table 1, we present a restriction-based prompt template consisting of four main parts: `[instruction]`, `[dialogue]`, `[restrictions]`, and `[answer]`. For each stage, we use different sentences for `[instruction]` and `[restrictions]`.

| Prompt Template |
| --- |
| `[instruction]` |
| Dialogue: `[dialogue]` |
| Restrictions: `[restrictions]` |
| `[answer]` |

Table 1: 🚫 **Restriction-based Prompt Template.**

## 2.4 STAGE 1: PREDICTING IMAGE-SHARING TURN

The goal of this stage is to predict all possible turns (e.g., $t - 3$, $t + 2$) that are likely to be appropriate to share an image on the next turn, together with speaker information (*who*) and rationale (*why*), given the entire dialogue history ($\{u_j\}_1^N$). For `[instruction]`, we ask LLMs to list the utterances in `[dialogue]` in descending order according to the confidence score (0-1) for the image-sharing turn. For `[dialogue]`, we use all utterances ($\{(u_j, s_j)\}_{j \neq t}^N$) with corresponding speakers in the given dialogue, excluding only the image-sharing turn $t$, to make LLM predict all possible turns that are originally not provided by the PhotoChat dataset. In addition, to make the `[dialogue]` more natural, we replace $s_j$ with Top-1K common names of US SSN applicants from 1990 to 2021 [1], followed by a previous work (Kim et al., 2022). For `[restrictions]`, we use three sentences; (1) your answer should be in the format of "<UTTERANCE> | <CONFIDENCE> | <SPEAKER> | <RATIONALE>", (2) you MUST select the utterance in the given dialogue, NOT generate a new utterance, (3) the rationale should be written starting with "To". Recent studies have

---

[1] https://catalog.data.gov/dataset/baby-names-from-social-security-card-applications-national-data

witnessed the success of LLMs in achieving encouraging performance on complex reasoning tasks, which benefited from the rationale by generating manually (Wei et al., 2022b; Wang et al., 2022) or automatically (Zhang et al., 2022b; Zelikman et al., 2022). Inspired by this, we also induce LLMs to generate rationale (*why*) together for high-quality generation results. For [answer], we allow LLMs to generate all possible image-sharing turns, formatted as lists (e.g., 1., 2.) following [restrictions] (i.e., *when*, *who*, *why*). Since the generated answer has a structured format, we parse LLM-generated answers by exploiting the regex pattern presented in the Appendix L. To find the image-sharing turn, we match the generated utterance among the set of utterances in a given dialogue by measuring the copy-and-paste ratio using token-level precision and recall score.

## 2.5 STAGE 2: GENERATING IMAGE DESCRIPTION

Without any help of visual foundation models (e.g., Stable Diffusion (Rombach et al., 2022), CLIP (Radford et al., 2021)), generating or retrieving various images relevant to the dialogue context by leveraging LLMs solely is challenging. To address this issue, we substitute the image retrieval task (introduced in PhotoChat) with the conditional text generation task. Therefore, this stage aims to generate an image description ($c_{t-3}$) relevant to the dialogue context at predicted image-sharing turns (e.g., $t-3$) from stage 1, given the previous dialogue ($\{(u_1, s_1), ..., (u_{t-3}, s_{t-3})\}$). For [dialogue], we put the previous dialogue $\{(u_j, s_j)\}_1^{t-3}$ and ([Sharing Image], $s_{t-2}$), as described in Figure 4 (left). For [restrictions], we use three sentences; (1) your answer should be written starting with "An image of" and in one sentence, (2) you do NOT include the speaker's name (i.e., [speaker1], [speaker2]) in the image description, (3) you should share a realistic image, NOT memes. Among these sentences, we provide an instruction - only depicting a realistic image, not a meme - into the LLMs because memes are not awkward in any turn. When you imagine memes, any phrases or sentences can be placed besides the images. For [answer], we allow LLMs to generate the image description by setting Image Description:.

## 3 EXPERIMENTS

### 3.1 EXPERIMENTAL SETUP

**Dataset.** We evaluate the *image-sharing* capabilities of LLMs using the test set from PhotoChat (Zang et al., 2021) dataset. This dataset, constructed via a crowdsourcing platform (i.e., Amazon Mechanical Turk), contains 10k multi-modal dialogues, each consisting of a set of non-consecutive turns. In our experiments, we use this set of non-consecutive turns as the [dialogue] in the input prompt to make the [dialogue] more natural and realistic.

**Language Models.** The primary objective is to assess the *image-sharing* capability of LLMs in terms of zero-shot performance, which necessitates complex reasoning. To achieve this, it is inevitable to leverage instruction-tuned large language models. For proprietary LLMs, we evaluate 8 models in total: 1) InstructGPT (Ouyang et al., 2022) with various model sizes (text-ada/babbage/curie/davinci-001/davinci-002 and text-davinci-003), 2) ChatGPT (OpenAI, 2023a), and 3) GPT-4 (OpenAI, 2023b). [2]. The text-davinci-001 and text-davinci-002 are derived from instruction-finetuning (IFT) and reinforcement learning from human feedback (RLHF), respectively, and both models have the same model size of 175B. For open-sourced LLMs, we evaluate 4 models in total: 1) VICUNA 13B (Chiang et al., 2023), 2) DOLLY 13B (Conover et al., 2023), 3) TULU 13B (Wang et al., 2023), and 4) LLAMA2 CHAT 13B (Touvron et al., 2023). We present the hyperparameter settings for each stage in the Appendix E.

**Evaluation Metrics.** To understand whether LLMs contain the *image-sharing* capability in terms of (1) *when to share* and (2) *what to share*, we compare LLM-generated results with the ground-truth (GT) image-sharing turn and corresponding annotated image from the PhotoChat dataset. In stage 1, we use PRECISION@K, RECALL@K, and F1@K to evaluate whether the GT image-sharing turn is among K possible image-sharing turns predicted from LLMs. In stage 2, we measure the quality of generated image descriptions at the GT turn using various metrics for Inter-Modal

---

[2]We conduct experiments with all language models by calling the OpenAI API between April-2023 and September-2023.

Consistency, Intra-Modal Consistency, and Diversity. For the inter-modal consistency, we measure the CLIPScore (Hessel et al., 2021) between the generated description and the image from the PhotoChat dataset. For the intra-modal consistency, we measure the relevance of the generated description to the dialogue context using a finetuned RoBERTa (Liu et al., 2019) model [3] on the dialogue contradiction detection (DECODE (Nie et al., 2020)) dataset. For measuring diversity, we use Dist-$n$ (Li et al., 2015; See et al., 2019) and Ent-$n$ (Han et al., 2022). In addition, we count the number of unique words, unique hypernyms, and Part-of-Speech (pos) tag words.

To evaluate the quality of LLM-generated results in stages 1 and 2, we compare three models (text-davinci-001/davinci-002/davinci-003) through human ratings. We randomly sample 100 dialogues where all three LLMs predict the same image-sharing turn for fair comparisons between LLMs. Three human raters assess each question based on four criteria: (1) image-sharing turn relevance, (2) image-sharing speaker adequacy, (3) image-sharing rationale relevance, and (4) image-caption relevance. All criteria, except for (2) (yes/no), employ a 4-point Likert scale. The Appendix G, H contains a detailed description of the questionnaires and system used for human evaluation.

## 4 EXPERIMENTAL RESULT

### 4.1 ZERO-SHOT PERFORMANCE

**Image-Sharing Turn Prediction.** As shown in Figure 2, overall, the zero-shot performance of LLMs improves as the scale of LLMs increases on F1@ALL metric, which indicates that a scaling law (Kaplan et al., 2020) also exists in the image-sharing turn prediction task. We mainly show that the InstructGPT (175B), ChatGPT, GPT-4 can share images to some extent by understanding the given dialogue context without additional training on the PhotoChat dataset. This result suggests that LLMs can be effective in social dialogues that require understanding and imagination of interactions between multiple people.

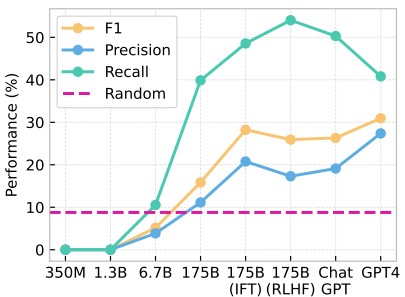

Figure 2: **Zero-Shot Performance.** We report the zero-shot performance when $k = all$. Every point represents an individual model. The pink dotted line denotes the random performance.

Interestingly, as shown in Figure 2, while the zero-shot performance of InstructGPTs ($< 175B$) is under the random performance, scaling the size of LLMs (i.e., text-davinci-001/davinci-002/davinci-003/ChatGPT, and GPT-4) significantly improves the zero-shot performance than the random one. Through this observation, we consider this *image-sharing* ability as an emergent ability (Wei et al., 2022a) in the zero-shot prompting, while the original work evaluates this ability in the few-shot prompting.

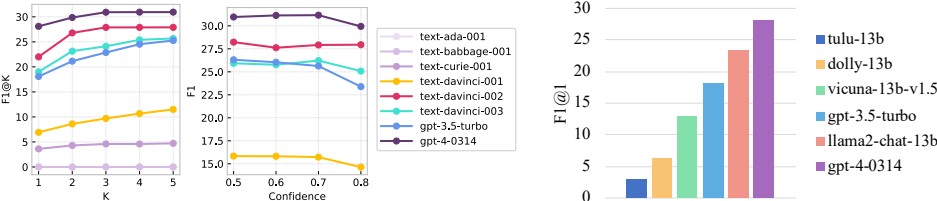

Figure 3: **Additional Results.** We show the effect of $k$ and confidence (left) and compare open-sourced LLMs with the proprietary LLMs (right).

**Effect of $k$.** As shown in Figure 3 (left), the F1@K performance for all LLMs improves consistently as $k$ increases. This trend is similar to the IR task where the performance increases when considering more relevant retrieved documents or images. Moreover, GPT-4 outperforms other LLMs across all different $k$.

**Effect of Confidence.** In Figure 3 (left), zero-shot performance diminishes with a confidence value of 0.8. This suggests that high-confidence image-sharing turns judged by LLMs might differ from

---

[3] https://huggingface.co/ynie/roberta-large_conv_contradiction_detector_v0

| Models | Inter-Modal Consistency | Intra-Modal Consistency | Diversity | | | | | | | | |
| | CLIPScore | NLI Score | Dist-1 | Dist-2 | Ent-1 | Ent-2 | # word | # hyp | # noun | # verb | # adj |
|---|---|---|---|---|---|---|---|---|---|---|---|
| Baseline | 0.6123 | - | 0.0927 | 0.1923 | 5.1699 | 7.9083 | 588 | 203 | 2728 | 333 | 338 |
| d-001 | 0.6156 | 0.9248 | 0.1266 | 0.4149 | 7.2714 | 10.8423 | 1644 | 535 | 3930 | 1244 | 795 |
| d-002 | 0.6341 | 0.9257 | 0.1303 | 0.4388 | 7.3687 | 11.0748 | 1633 | 597 | 3860 | 1009 | 913 |
| d-003 | 0.6387 | **0.935** | 0.1427 | 0.4902 | 7.9422 | 11.7782 | 2396 | 790 | 5062 | 1750 | 1373 |
| ChatGPT | 0.623 | 0.9265 | **0.1588** | **0.5096** | 8.0953 | 11.7731 | 2470 | 804 | 4970 | 1510 | 1407 |
| GPT-4 | **0.6566** | **0.935** | 0.1318 | 0.4676 | **8.1983** | **12.021** | **3084** | **1033** | **6747** | **2610** | **2315** |

Table 2: **Results of Stage 2.** We report the quality of generated image descriptions from LLMs on various metrics. For calculating CLIPScore Hessel et al. (2021), we leverage the CLIP-base model (ViT-B/32). The baseline denotes the performance of PhotoChat Zang et al. (2021) dataset. d-001,002, and d-003 denote text-davinci-001, 002, and text-davinci-003, respectively.

human choices. The reliability of the model-generated confidence value may also contribute to this performance drop.

**Open-Sourced LLMs.** We compare open-sourced LLMs with proprietary LLMs (e.g., ChatGPT, GPT-4) regarding the image-sharing capability. As shown in Figure 3 (right), GPT-4 outperforms the other LLMs. Notably, LLAMA2-CHAT-13B surpasses ChatGPT, indicating that high-quality datasets and optimization for dialogue use cases may enhance image-sharing ability.

**Image Description Generation.** Table 2 compares the quality of image descriptions generated by LLMs on various metrics to the PhotoChat dataset. Overall, all LLMs outperform the baseline regarding inter-modal consistency and diversity, suggesting they can produce more specific and diverse descriptions related to the given dialogue context. As shown in Figure 4, all LLMs generate image descriptions that are semantically relevant to the provided dialogue. Particular;y, RLHF-equipped LLMs, such as text-davinci-003, ChatGPT, and GPT-4, tend to generate lengthy image descriptions that depict specific situations or objects, for instance, "*sitting*", "*next to them*", "*Coke Zero, Cherry Coke, and Vanilla Coke*". This implies that LLMs with RLHF can simulate human visual imagination ability (Lu et al., 2022; Zhu et al., 2023; Lu et al., 2023), benefiting from learning with human feedback.

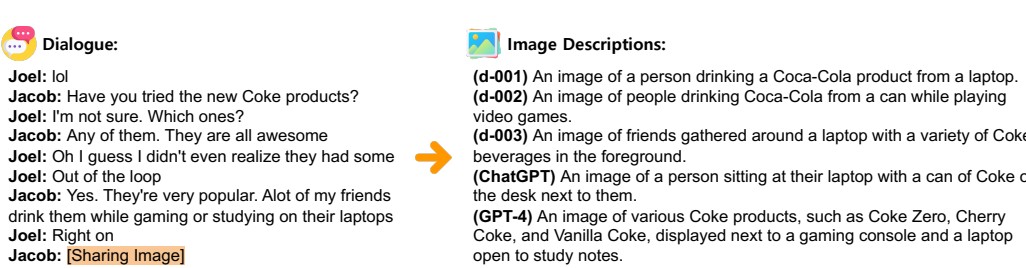

Figure 4: **LLM-Generated Image Description Comparisons.** We present five image descriptions (right) generated by LLMs given the same dialogue (left) with the image-sharing turn (i.e., [Sharing Image]).

## 4.2 HUMAN EVALUATION

This work aims to test the *image-sharing* capabilities of LLM in the zero-shot setting by (1) predicting all potential image-sharing turns and (2) generating relevant image descriptions at the predicted turns. Accordingly, we conduct the human evaluation to determine the relevance of the generated image-sharing turns, speakers, rationales, and image descriptions, focusing on text-davinci-001, text-davinci-002, and text-davinci-003. We achieve an average score of 3.04 for turn relevance and 1.66 for speaker ade-

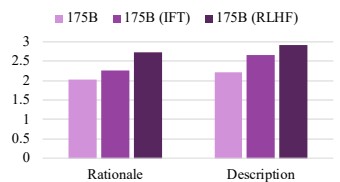

Figure 5: **Human Evaluation Performance.** We present the results of the human evaluation in terms of both rationale relevance and description relevance.

quacy, indicating that the results generated by LLMs are generally favorable to human evaluators. Furthermore, we compare the rationales and descriptions generated by each LLM when provided with

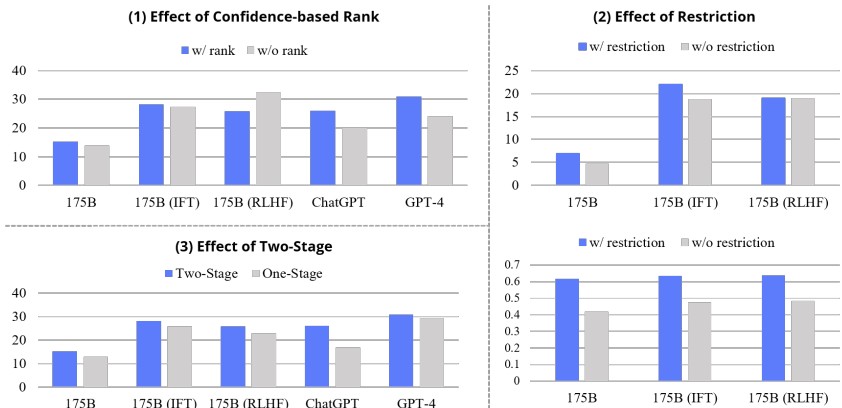

Figure 6: **Ablation Studies.** We conduct three different experiments and report the F1 performance.

the same dialogue. As depicted in Figure 5, humans prefer the rationales and descriptions generated by LLM with RLHF over those generated by other LLMs.

## 4.3 ABLATION STUDIES

**Prompting without Confidence-based Ranking.** In stage 1, we prompt LLMs to generate the image-sharing turns based on confidence-based ranking. To investigate the effect of this ranking, we modified the prompt by removing the corresponding part from the [instruction]. As depicted in Figure 6, prompts based on ranking generally lead to improved zero-shot performance across almost all LLMs, except for text-davinci-003. This result suggests that the ranking-based prompt predicts image-sharing turns grounded in confidence, rather than merely following the chronological order of utterances in the provided dialogue, which is the predominant behavior in the non-ranking-based prompt method.

**Prompting without Restrictions.** In stage 1, comparable zero-shot performance is partially due to the beneficial effects of the Restrictions: token, which effectively guides LLMs to adhere to the provided instructions strictly. To evaluate the impact of Restrictions:, we modify the prompt by placing the three sentences previously under Restrictions: immediately after the instruction. The revised prompt template is provided in the Appendix J.2. Then, we provide this modified prompt to LLMs in both stages and measure F1@1 and CLIPScore on the PhotoChat dataset. As shown in Figure 6, eliminating Restrictions: leads to a decrease in performance in both stages relative to the original results (highlighted in blue in Figure 6). These findings suggest that the *image-sharing* capabilities require complex reasoning.

**Two-Stage vs. One-Stage.** We conduct an additional experiment to determine whether the framework could operate in a single phase. For this experiment, we designed a prompt by seamlessly merging the two prompts in the proposed two-stage framework. The results are depicted in Figure 6. We noticed a decline in the F1@ALL metric for image-sharing capability. We believe that asking the LLM to predict the image-sharing turn while also generating a pertinent image description might have exerted a substantial load on the LLMs. From this observation, we can deduce that while the single-phase approach showcases the LLM's image-sharing ability, a two-phase method might be preferable for superior performance. Nevertheless, the drawback of the two-phase approach is that it would be twice as expensive each time we invoke models like closed LLMs (e.g., OpenAI's GPT, Google Bard, Anthropic Claude, etc.) via their APIs.

| Models | # U. | Avg. # U./D. |
|---|---|---|
| ada-001 | 19 | 1.583 |
| babbage-001 | 34 | 1.789 |
| curie-001 | 2001 | 2.308 |
| davinci-001 | 6523 | 6.795 |
| davinci-002 | 2730 | 2.82 |
| davinci-003 | 2185 | 2.262 |
| ChatGPT | 3960 | 4.091 |
| GPT-4 | 3076 | 3.178 |

Table 3: **Statistics of Image-Sharing Turns.** U./D. denotes predicted turns by a dialogue.

Given this cost implication, opting for open-sourced LLMs (e.g., LLAMA2-CHAT) might present a feasible alternative, as reported in Figure 3.

### 4.4 ANALYSIS OF LLM-GENERATED RESULTS

**Statistics of Image-Sharing Turns.** Table 3 presents the statistics of image-sharing turns generated by LLMs. On average, LLMs generate roughly 3.1 image-sharing turns per dialogue, a reasonable count compared to the 2.59 turns in a multi-modal dialogue dataset (Feng et al., 2022) collected from social media. Notably, the text-davinci-001 model generates significantly more image-sharing turns than other LLMs.

**Diversity Comparisons.** In Table 2, we compare the diversity of generated image descriptions with the number of unique words and hypernyms from WordNet (Miller, 1995). Compared to the baseline, the LLM-generated description includes roughly 3.3× more hypernyms; 3.4× more words, indicating that LLM generates diverse descriptions covering more variety of open-domain topics. In addition, Table 2 shows the linguistic diversity by counting the number of part-of-speech (POS) tags using the en_core_web_sm POS tagger from spaCy (Honnibal et al., 2020). Similar to the above result, the generated description contains 1.5× more nouns, 4.3× verbs, 3.4× adjectives, and 31.0× adverb words than the baseline. Especially, LLMs with RLHF tend to generate more specific descriptions with more use of words than other LLMs.

**Rationale Distribution.** To better understand the generated rationales, we perform an analysis of their verb-noun structures. Using spaCy (Honnibal et al., 2020), we parse the rationales and extract the root verb along with its first direct noun object. Given the restriction in the prompt, a rationale should start with "To." Hence, we solely consider rationales with the "To verb noun" structure during this analysis. Out of a total of 4,320 rationales generated by the text-davinci-002 and text-davinci-003 models, 3,732 rationales adhere to this structure, whereas 588 rationales contain more complex clauses (e.g., *To check if a person can see the photo.*).

During the analysis of the generated rationales, we observe that the verb "provide" is the most frequently used in the rationales, typically in conjunction with "information", "context", and "representation". This suggests that the generated rationales often aim to supply relevant information or context. The verbs "show" and "share" are also frequently used, indicating a communicative intent within the rationales. Lastly, the verb "express" is used primarily with emotional or affective states such as "interest", "reaction", "excitement", and "appreciation". We present the complete breakdown table in the Appendix.

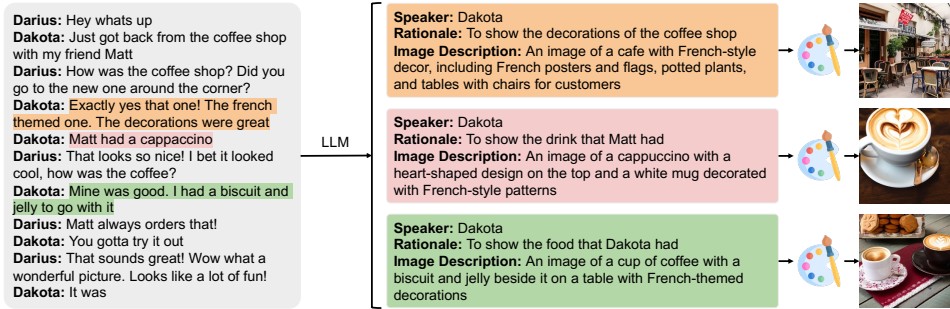

Figure 7: **An example of PhotoChat++.** Followed by our framework, we construct the PhotoChat++ by generating appropriate images using Stable Diffusion (🎨) by prompting predicted image descriptions from LLM (i.e., text-davinci-003), which are highlighted in orange, pink, and green boxes.

### 4.5 PHOTOCHAT++

As reported in prior works (Yoo et al., 2021; West et al., 2021; Lee et al., 2021; Kim et al., 2022; Chun et al., 2022), machine-annotated or -generated datasets effectively enhance generalization performance. Inspired by these works, we augment the PhotoChat dataset with LLM-generated image-sharing turns (as seen with text-davinci-003 in Table 3) and corresponding images produced by Stable Diffusion (Rombach et al., 2022) [4], resulting in PhotoChat++. We implement simple text and image retrieval models to examine if PhotoChat++ improves generalization performance on unseen multi-modal dialogue datasets (detailed in the Appendix F). We train these models on both PhotoChat

---

[4] https://huggingface.co/stabilityai/stable-diffusion-2-1-base

| Eval → | PhotoChat | | | | PhotoChat++ | | | | MMDialog | | | |
|---|---|---|---|---|---|---|---|---|---|---|---|---|
| Train ↓ | R@1 | R@5 | R@10 | MRR | R@1 | R@5 | R@10 | MRR | R@1 | R@5 | R@10 | MRR |
| *Image Retrieval* | | | | | | | | | | | | |
| PhotoChat | $16.51_{\pm0.27}$ | $43.37_{\pm0.82}$ | $60.44_{\pm1.59}$ | $29.91_{\pm0.38}$ | $14.6_{\pm0.95}$ | $39.67_{\pm0.5}$ | $56.09_{\pm0.55}$ | $27.52_{\pm0.64}$ | $5.88_{\pm0.21}$ | $19.21_{\pm0.92}$ | $29.95_{\pm1.3}$ | $14.13_{\pm0.49}$ |
| PhotoChat++ | $\mathbf{16.92}_{\pm1.18}$ | $\mathbf{44.89}_{\pm1.18}$ | $\mathbf{61.77}_{\pm1.05}$ | $\mathbf{30.85}_{\pm0.81}$ | $\mathbf{21.64}_{\pm0.32}$ | $\mathbf{53.69}_{\pm1.02}$ | $\mathbf{69.75}_{\pm0.27}$ | $\mathbf{36.63}_{\pm0.38}$ | $\mathbf{8.25}_{\pm0.47}$ | $\mathbf{24.95}_{\pm1.21}$ | $\mathbf{37.0}_{\pm1.65}$ | $\mathbf{17.82}_{\pm0.76}$ |
| *Next Response Prediction* | | | | | | | | | | | | |
| PhotoChat | $6.02_{\pm0.26}$ | $19.81_{\pm0.78}$ | $31.67_{\pm1.68}$ | $14.72_{\pm0.45}$ | $4.56_{\pm0.39}$ | $17.6_{\pm0.46}$ | $28.3_{\pm1.19}$ | $12.7_{\pm0.43}$ | $2.34_{\pm0.24}$ | $9.27_{\pm0.89}$ | $16.22_{\pm1.25}$ | $7.92_{\pm0.54}$ |
| PhotoChat++ | $\mathbf{6.43}_{\pm0.93}$ | $\mathbf{23.06}_{\pm1.52}$ | $\mathbf{34.24}_{\pm1.15}$ | $\mathbf{15.9}_{\pm0.85}$ | $\mathbf{7.03}_{\pm0.49}$ | $\mathbf{24.06}_{\pm0.93}$ | $\mathbf{37.0}_{\pm1.16}$ | $\mathbf{16.81}_{\pm0.59}$ | $\mathbf{2.67}_{\pm0.07}$ | $\mathbf{9.86}_{\pm0.26}$ | $\mathbf{16.29}_{\pm0.38}$ | $\mathbf{8.19}_{\pm0.15}$ |

Table 4: **Text and Image Retrieval Performance.** We report the text and image retrieval performance across five runs on three multi-modal dialogue datasets: PhotoChat, PhotoChat++, and MMDialog.

and PhotoChat++ and evaluate them on PhotoChat, PhotoChat++, and MMDialog (Feng et al., 2022) using an unseen setting. Table 4 reveals that models trained on the PhotoChat++ dataset outperform on all three multi-modal dialogue datasets. Notably, in MMDialog, there's a significant performance boost with PhotoChat++, underscoring the effectiveness of our two-stage framework in constructing PhotoChat++. We present an example of PhotoChat++, as shown in Figure 7.

## 5 RELATED WORK

**Multi-Modal Dialogue Dataset.** Existing studies predominantly fall into two categories, depending on whether the image in the dialogue is *grounded* or *sharing*. Image-grounded dialogue tasks are designed to answer questions (Antol et al., 2015; Das et al., 2017; Kottur et al., 2019) or generate natural conversations (Mostafazadeh et al., 2017; Shuster et al., 2018; Meng et al., 2020; Wang et al., 2021b; Zheng et al., 2021) about given images. Nevertheless, it is common to share images pertinent to dialogue contexts in everyday conversations for the purpose of reinforcing social bonding, as well as enhancing engagement and interest. Recent studies have proposed datasets that encapsulate this image-sharing behavior. This has been achieved by collecting a human-human dialogue dataset (PhotoChat) via crowdsourcing (Zang et al., 2021), a large-scale dataset (MMDialog) from social media (Feng et al., 2022), or constructing datasets automatically using vision-and-language models (Lee et al., 2021). In this work, our focus is exclusively on the PhotoChat dataset to gain a deeper understanding of the *image-sharing* capabilities of LLMs. We do not include automatically constructed datasets or the MMDialog due to the considerable expense associated with conducting experiments using LLMs.

**Prompting Large Language Models.** Recent studies have witnessed the success of large language models, such as GPT-3 (Brown et al., 2020), Instruct GPT-3 (Ouyang et al., 2022), ChatGPT (OpenAI, 2023a), GPT-4 (OpenAI, 2023b), in a zero-/few-shot performance, which benefited from pretrained on a massive amount of corpus with instruction-finetuning (IFT) and reinforcement learning from human feedback (RLHF). The use of these models, in conjunction with "prompt engineering," has unlocked the abilities of LLMs, even emergent ones (Wei et al., 2022a), across various tasks. These tasks range from dialogues (Lee et al., 2022a; Kim et al., 2022; Lee et al., 2022b), complex reasoning tasks (Wei et al., 2022b; Kojima et al., 2022), and theory-of-mind (ToM)(Sap et al., 2022; Kosinski, 2023), to image classification(Yang et al., 2022; Pratt et al., 2022; Menon & Vondrick, 2022; Zhang et al., 2023) and multi-modality (Lu et al., 2023; Han et al., 2023). In this work, we assess the *image-sharing* capabilities of LLMs in a zero-shot setting, which is the first study.

## 6 CONCLUSION

In this paper, we explore the *image-sharing* capabilities of LLMs in a zero-shot prompting by introducing a two-stage framework and a restriction-based prompt template. Our extensive experiments demonstrate the effectiveness of our restriction-based prompt template in enhancing zero-shot performance across both stages, with GPT-4 achieving state-of-the-art performance. We also reveal that the *image-sharing* ability is an emergent ability in the zero-shot prompting. In a comprehensive analysis, we observe that LLMs can generate specific and diverse image descriptions. Moreover, we augment the PhotoChat dataset, namely PhotoChat++, which enhance the generalization performance. In future works, we will assess this capability in a few-shot setting using additional multi-modal dialogue datasets.

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

## A  BROADER IMPACTS

We report several potential issues with our proposed framework. First, generated image descriptions may propagate social bias because GPT-3 can still produce harmful content, including social bias and offensiveness(Baheti et al., 2021; Hartvigsen et al., 2022). Second, this issue has resulted in the inclusion of problematic descriptions in the constructed PhotoChat++ dataset, leading to socially-biased images generated using Stable Diffusion (Rombach et al., 2022). As a result, when vision-and-language models like CLIP (Radford et al., 2021) and DALL-E (Ramesh et al., 2021) are trained on this augmented dataset, they may exhibit social biases. As reported in (Wang et al., 2021a), even if we give the gender-neutral query to CLIP (Radford et al., 2021) model, the CLIP model sometimes retrieves images causing gender-bias issues. We are concerned that this problematic issue may exist in the augmented dataset. Therefore, the image retrieval model trained on this dataset may sometimes retrieve biased images. In addition, text-to-image generative models learn social biases from the augmented dataset, as reported in the prior work (Cho et al., 2022). We should consider this problem important when building a multimodal search model.

## B  LIMITATIONS

Here, we highlight some limitations of our work. Firstly, our restriction-based prompt template is rather lengthy, which complicates expansion into the few-shot setting. We anticipate that conducting few-shot prompting to utilize the image-sharing capability of LLMs would result in better performance compared to zero-shot prompting. Secondly, LLMs tend to over-generate image descriptions even in the absence of specific demographic information such as age or appearance. For instance, in the description, "An image of a woman with long, brown hair wearing a flowy white dress and brown boots," there is no reference to long hair in the given dialogue. Providing additional information (e.g., persona) can enhance the relevance of image descriptions generated by LLMs.

## C  DISCUSSIONS

**Towards Better Image-Sharing Ability.**   As shown in our experiment, the likelihood of performance improvement is high as the model's size increases or when it is trained with alignment to human preference. This suggests that the image-sharing ability is subjective and resembles human-like tasks. Therefore, receiving real-time feedback through interactive mode (a form of human-AI collaboration) and further training the model using the RLHF method could lead to better performance, aligning the model's actions favorably with image-sharing ability.

Furthermore, understanding conversational context is essential, and imbuing the model with the ability of perspective-taking, understanding situations from the user's point of view, could lead to performance enhancement. For instance, when a user is feeling down due to poor test results, the model could not only provide empathy through text but also share a picture of a dog based on the user's fondness for dogs and the current context of struggling with test scores, thereby offering multi-faceted empathy.

In addition, unlike image-grounded dialogue, image-sharing scenarios might lack explicit information from previous conversations. For instance, understanding what "it" refers to in "I love it" requires considering the preceding conversational context. Thus, it's important to consider coreference resolution. Moreover, while sharing images, incorporating information about significant utterances from previous dialogues or using keywords and keyphrases could likely improve performance.

As depicted in Figure 7's orange-generated results, the language model might sometimes over-generate due to excessive creativity. For instance, if the conversation only contains information about a coffee shop without mentioning "French-style," the model might still produce the word "French." Such cases could pose challenges in practical applications where inappropriate images could be retrieved.

In practical applications, it's beneficial to consider the user's satisfaction and share images that account for their personal information. For example, if a user mentions, "I work in a hotdog stand," and their friend, who also works there, has a picture related to selling hotdogs in their phone album, it would be more suitable to share an image depicting the user themselves selling hotdogs rather than

an image with the friend. Of course, obtaining explicit consent for sharing personal information is crucial.

Additionally, beyond improving the image-sharing ability, at the application level, using videos could enhance user engagement. Exploring this avenue could be a promising direction for future research.

**Intrinsic Properties of LLMs.** We believe that the intrinsic properties of LLM, which have been experimentally proven in various studies, have influenced image-sharing ability.

- **Understanding the dialogue context:** It's essential to grasp the conversation topic holistically, emotional shifts between users, and general knowledge. Recent research results have shown that language models possess these abilities.

- **Understanding the interlocutor's mental state:** It is important to comprehend the interlocutor is situation to determine whether sharing an image is appropriate. For instance, if the interlocutor is upset, it might be better to respond empathetically rather than share an image. This ability is highly related to the Theory-of-Mind (ToM). Recently, LLMs have achieved competitive performance in Theory-of-Mind (ToM) tasks, which may influence image-sharing ability.

- **Understanding the intent:** From the model's perspective, sharing an image can be seen as intent. Many language models have demonstrated good performance in task-oriented dialogue tasks.

- **Visual imagination ability:** To share an appropriate image, one must imagine which image is best. This capability has been empirically proven in various recent studies. We investigated the C4 dataset, a representative pretraining dataset for LLMs, to analyze why this capability is manifested. The data discovered in C4 consists of pairs of images and their corresponding captions. These captions contain words/phrases related to visual imagination ability, such as "depict" and "photo of." Moreover, on blogs, images often appear consecutively along with stories. Due to these elements, the LLM learned an inherent visual, and imaginative capability during its pretraining phase.

**Image Retrieval for Real Use Cases.** We believe that utilizing a text-to-image retrieval model (e.g., CLIP (Radford et al., 2021)) to build an image-suggesting system would be more effective in actual use cases. The image generation step mentioned in our paper is used for creating the PhotoChat++ dataset, an application of our proposed framework, by generating images using StableDiffusion. Therefore, at a practical level, within our framework's stage 2, the LLM-generated image description can be used for text-to-image retrieval, allowing for the sharing of more diverse and up-to-date images from phone albums or the internet.

## D    MOTIVATION BEHIND PROVIDING FULL CONTEXT IN STAGE 1

In Stage 1, the full history is given as input. In Stage 2, only the previous conversation context is provided based on the image-sharing turn identified in Stage 1 (as depicted on the left side of Figure 4). Due to differing objectives, we allowed the LLM to view the entire future dialogue in Stage 1. This paper argues that multiple turns can serve as image-sharing turns (as detailed in Section 2). Therefore, we needed to identify other possible turns in the original PhotoChat dataset, which only had a single image-sharing turn. Given that the dialogue context was already fixed, it was crucial to find image-sharing turns that wouldn't disrupt the existing conversation flow after inserting relevant images at the image-sharing turns. Suppose we provided input to the LLM at every turn, similarly at Stage 2. In that case, there might be inconsistencies between the dialogue coherence after the image sharing and the subsequent dialogue content (which existed in the original PhotoChat). Thus, while giving the LLM the future dialogue as input can act as a hint, it was an inevitable choice in our experiments to validate our claim.

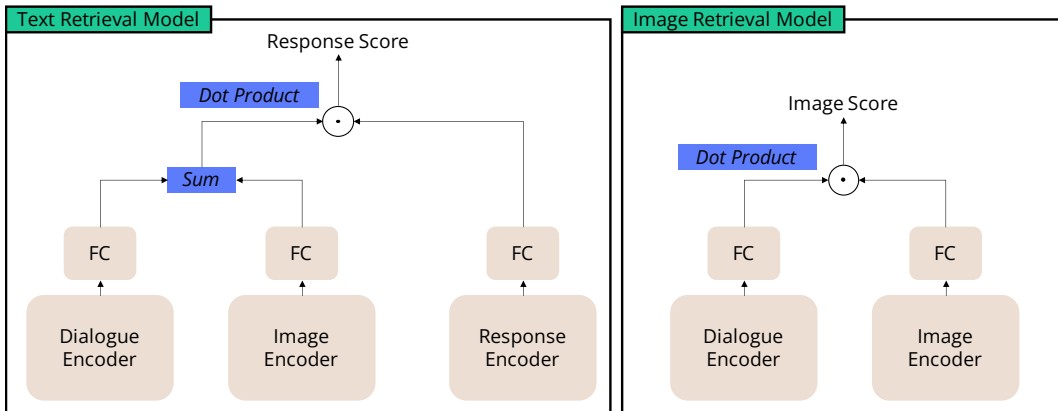

Figure 8: Architectures of two baseline models: Text retrieval and Image retrieval.

# E IMPLEMENTATION DETAILS OF LLMS

To evaluate the *image-sharing* capabilities of LLMs, we call InstructGPT (Ouyang et al., 2022), ChatGPT (OpenAI, 2023a), and GPT-4 (OpenAI, 2023b) by calling OpenAI API. All experiments are conducted on two A100 (40GB) GPU. For each stage, the generation configuration is as follows:

- For Stage 1, we set maximum tokens to 1024, temperature to 0.0, frequency penalty to 0.0, presence penalty to 0.0, top_p to 1.0, and stop tokens to \n\n.
- For Stage 2, we set maximum tokens to 1024, temperature to 0.9, frequency penalty to 0.0, presence penalty to 0.4, top_p to 0.95, and stop tokens to a default setting.

# F DETAILS OF EXPERIMENTAL SETTINGS

To explore how our dataset affects both text and image retrieval tasks, we implement two simple and standard baseline retrieval models for text-to-image and image-to-text settings.

## F.1 TASK DEFINITION

Follwing (Lee et al., 2021; Zang et al., 2021), we explain the formulation of two main tasks - next response prediction and image retrieval. Let us assume that we have a multi-modal dialogue $\mathcal{D} = \{(u_j, i_j, c_j)\}_1^N$ where $N$ denotes the number of dialogue turns, and $j = t$ is the turn that an image sharing behavior occurs. Then, each task is formulated as follows.

**Next response prediction** is to predict the next utterance at turn $t + 1$ given the dialogue history $(\{u_j\}_1^t)$ and image $i_t$.

**Image retrieval** is to retrieve relevant image at turn $t$ given the dialogue history $(\{u_j\}_1^{t-1})$.

Following (Shuster et al., 2018; Lee et al., 2021), we set the number of retrieval candidates to 100 and use Recall@{1,5,10} and mean reciprocal rank (MRR) for the evaluation metrics.

## F.2 BASELINE MODELS

As illustrated in Figure 8, we present the architecture of baseline models: the text retrieval and image retrieval models. We provide a detailed description of baseline models below.

**Text Retrieval Model.** The text retrieval model comprises three main components: the dialogue encoder, the response encoder, and the image encoder. The dialogue encoder processes the entire dialogue history into a fixed-size representation. To achieve this, we use the BERT model (Devlin

et al., 2018). The dialogue history consists of up to three turns preceding the current turn. Each turn is concatenated using the `[SEP]` special token. The response encoder is responsible for converting the response into a fixed-size representation. While it also utilizes the BERT model, the specific BERT version used here differs from that employed in the dialogue encoder. After processing the text with BERT, we apply mean pooling to the text representations for both the dialogue and response encoders. The pooled representations are passed through a linear projection layer, followed by the ReLU activation function (Nair & Hinton, 2010). The image encoder is to extract feature vectors from images, and for this purpose, we utilize the CLIP-base model (Radford et al., 2021). Once the feature vectors are extracted from the dialogue and images, we perform an element-wise addition of the image vectors and dialogue vectors. We calculate the dot product between the response feature vector and the resulting summed vector to compute the loss.

**Image Retrieval Model.** The image retrieval model comprises two main components: the dialogue encoder and the image encoder. The dialogue encoder utilizes the BERT-base model to transform the dialogue into a representation. After encoding, we apply mean pooling to the text representations derived from this dialogue encoder. For image representation, we employ the CLIP-base model. Following the encoding processes, the image and dialogue vectors are passed through separate linear projection layers, each followed by a ReLU activation function. We calculate the dot product between the image feature vector and the dialogue vector to determine the loss.

### F.3 IMPLEMENTATION DETAILS

We implement baseline models based on PyTorch Lightning. All experiments are conducted on two A100 GPUs (40GB). To accelerate the training time, we apply distributed training to baselines. We follow the hyperparameter settings similar to the previous works (Lee et al., 2021; Zang et al., 2021), which are described as follows:

**Text retrieval.** In our experiment, we set the batch size to 256, the learning rate to 5e-5, and the gradient clipping value to 2.0. We use the AdamW optimizer with a cosine learning rate scheduler. We set the warm-up ratio as 0.1% and weight decay as 0.2.

**Image retrieval.** We set the batch size to 256. We also use the AdamW optimizer with an initial learning rate of 2e-5 and decaying 0.1%

**Training.** Since our dataset contains several images per utterance, we randomly choose one image in each batch. We do not update the parameter of the image encoder.

## G   HUMAN EVALUATION QUESTIONNAIRE

We present a list of questions and multiple-choice options used for the human evaluation.

- Image-Sharing Turn Relevance: Do you think the image-sharing turn in the given dialogue is appropriate?

  **Options:** 1: Not at all / 2: A little / 3: Somewhat / 4: A lot

- Image-Sharing Speaker Adequacy: Do you think the speaker who shared the image in the given dialogue is appropriate? (When you selected 3 or 4 in Image-Sharing Turn Relevance)

  **Options:** 1: No / 2: Yes

- Image-Sharing Rationale Relevance: Do you think the reason for sharing the image in the given dialogue is valid? (When you selected 3 or 4 in Image-Sharing Turn Relevance)

  **Options:** 1: Not at all / 2: A little / 3: Somewhat / 4: A lot

- Image Description Relevance: Given the dialogue context, how relevant do you think the image description is? (When you selected 3 or 4 in Image-Sharing Turn Relevance)

  **Options:** 1: Not at all / 2: A little / 3: Somewhat / 4: A lot

## H HUMAN EVALUATION SYSTEM

As shown in Figure 9, we present a screenshot of human evaluation system. We implement the human evaluation system using Label Studio (Tkachenko et al., 2020-2022).

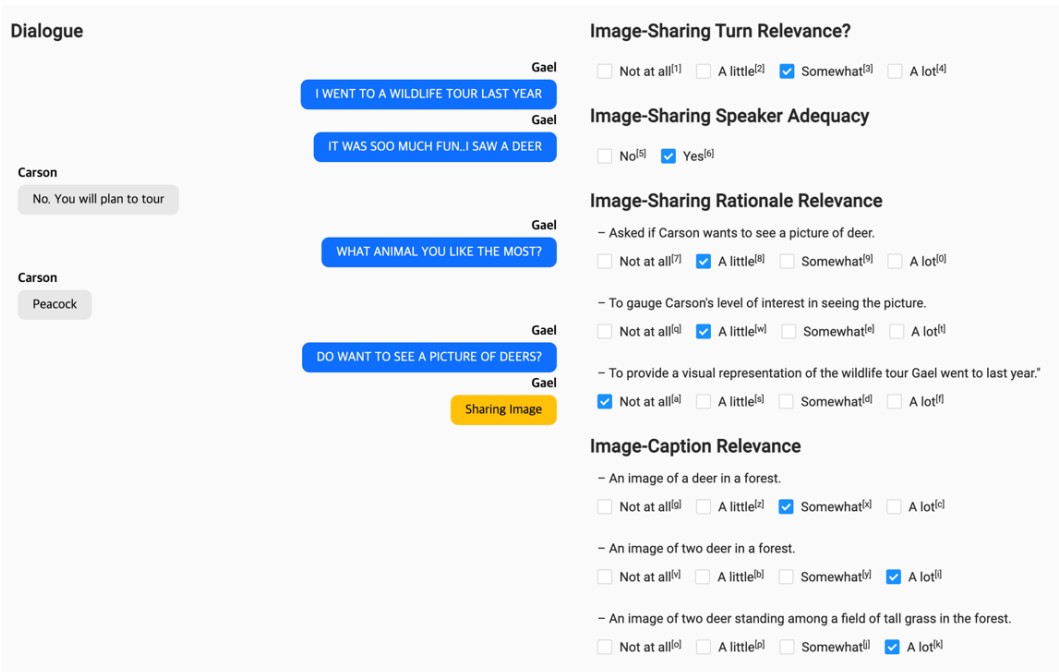

Figure 9: **Screenshot of the human evaluation system.**

## I    RATIONALE DISTRIBUTION

We present the rationale distribution as shown in Table 5.

| Verb | Object | Count | Example |
|---|---|---|---|
| provide | information | 612 | To provide more information about the moth she saw. |
| | context | 445 | To provide context for the conversation. |
| | representation | 397 | To provide a visual representation of the beverage person is talking about. |
| | evidence | 215 | To provide visual evidence of the fun time they had together. |
| show | interest | 174 | To show his interest in seeing the photo. |
| | image | 173 | To show the image of the letters he formed with the dough. |
| | person | 149 | To show person that he is okay with the weather. |
| | audience | 111 | To show the audience the fun person is having on his vacation. |
| share | image | 145 | To share the image of the birthday party. |
| | photo | 13 | To share the photo with person. |
| express | interest | 30 | To express interest in person's story |
| | reaction | 27 | To express her reaction to the image. |
| | excitement | 17 | To express excitement about the workshop. |
| | appreciation | 12 | To express her appreciation for the cake. |
| invite | person | 38 | To invite person to see the picture of the table. |
| ask | person | 25 | To ask person to share an image of his recent cooking. |
| | question | 6 | To ask a follow-up question about the image. |
| encourage | person | 23 | To encourage person to share his most memorable dinner. |
| introduce | image | 12 | To introduce the image. |
| | topic | 8 | To introduce the topic of the conversation. |
| gauge | interest | 18 | To gauge person's interest in the baked goods. |
| give | opportunity | 13 | To give person the opportunity to see a photo of Hannah. |
| engage | person | 9 | To engage person in the conversation and to show her the photo Zora sent. |
| emphasize | importance | 8 | To emphasize the importance of spending time with kids. |
| indicate | interest | 7 | To indicate person's interest in seeing the photo. |

Table 5: **Rationale Distribution.** The top 20 most common root verbs and their up to 4 direct noun objects in the generated rationale. Only pairs with a count of 5 or more are included.

## J    PROMPT TEMPLATES

Here, we present all prompt templates used in our work, such as restriction-based prompt templates for each stage, and several prompt templates for the ablation studies.

### J.1    RESTRICTION-BASED PROMPT TEMPLATES

We present restriction-based prompt templates for each stage in our proposed framework, as shown in Figure 10.

### J.2    PROMPT TEMPLATES FOR ABLATION STUDIES

We present prompt templates used in our ablation studies, as shown in Figure 11, Figure 12, Figure 13, and Figure 14.

## K    MORE EXAMPLES OF PHOTOCHAT++

We provide more examples of PhotoChat++ dataset.

**Prompt Template for Stage 1:**
The following is a dialogue between `[speaker1]` and `[speaker2]`. You should share an image to make the following dialogue more interesting and engaging. The dialogue is provided line-by-line. In the given dialogue, select all utterances that are appropriate for sharing the image in the next turn, and write the speaker who will share the image after the selected utterance. You should also provide a rationale for your decision. Please list the selected utterances in descending order according to the confidence (0∼1) of your choice of how appropriate the utterance is for sharing an image now.

Dialogue:
`[dialogue]`

Restrictions:
(1) your answer should be in the format of "<UTTERANCE> | <CONFIDENCE> | <SPEAKER> | <RATIONALE>".
(2) you MUST select the utterance in the given dialogue, NOT generate a new utterance.
(3) the rationale should be written starting with "To".

Answer:
1.

- - - - - - - - - - - - - - - - - - - - - - - - - - - - - - - - - - - - - - - - - - - - - - - - - - - - - - - - -

**Prompt Template for Stage 2:**
The following is a dialogue between `[speaker1]` and `[speaker2]`. The dialogue is provided line-by-line. `[speaker1]` shares an image in a given dialogue to make the following dialogue more interesting and engaging, marked in [Sharing Image]. Depict the most appropriate image to be shared in the next turn, in detail.

Dialogue:
`[dialogue]`

Restrictions:
(1) your answer should be written starting with "An image of" and in one sentence.
(2) you do NOT include the speaker's name (i.e., `[speaker1]`, `[speaker2]`) in the image description.
(3) you should share a realistic image, NOT memes.

Image Description:

Figure 10: **Prompt Templates for Image-Sharing Behavior.** A prompt template for stage 1 (**top**). A prompt template for stage 2 (**bottom**).

**Prompt Template for Stage 1 (w/o Restriction):**
The following is a dialogue between `[speaker1]` and `[speaker2]`. You should share an image to make the following dialogue more interesting and engaging. The dialogue is provided line-by-line. In the given dialogue, select all utterances that are appropriate for sharing the image in the next turn, and write the speaker who will share the image after the selected utterance. Please list the selected utterances in descending order according to the confidence (0~1) of your choice of how appropriate the utterance is for sharing an image now. You should also provide a rationale for your decision, starting with "To". Your answer should be in the format of "<UTTERANCE> | <CONFIDENCE> | <SPEAKER> | <RATIONALE>". You MUST select the utterance in the given dialogue, NOT generate a new utterance.

Dialogue:
`[dialogue]`

Answer:
1.

Figure 11: **Prompt Templates for Stage 1 (w/o Restriction).** A prompt template used in the ablation study in Stage 1 for validating the effect of restriction.

**Prompt Template for Stage 2 (w/o Restriction):**
The following is a dialogue between `[speaker1]` and `[speaker2]`. The dialogue is provided line-by-line. `[speaker1]` shares an image in a given dialogue to make the following dialogue more interesting and engaging, marked in [Sharing Image]. Depict the most appropriate image to be shared in the next turn, in detail. Your answer should be written starting with "An image of" and in one sentence. You do NOT include the speakers' name (i.e., `[speaker1]`, `[speaker2]`) in the image description. You should share a realistic image, NOT memes.

Dialogue:
`[dialogue]`

Image Description:

Figure 12: **Prompt Templates for Stage 2 (w/o Restriction).** A prompt template used in the ablation study in Stage 2 for validating the effect of restriction.

**Prompt Template for One-Stage:**

The following is a dialogue between `[speaker1]` and `[speaker2]`. You should share an image to make the following dialogue more interesting and engaging. The dialogue is provided line-by-line. In the given dialogue, select all utterances that are appropriate for sharing the image in the next turn, and write the speaker who will share the image after the selected utterance. You should also provide a rationale for your decision and describe the relevant image. Please list the selected utterances in descending order according to the confidence (0∼1) of your choice of how appropriate the utterance is for sharing an image now.

Dialogue:
`[dialogue]`

Restrictions:
(1) your answer should be in the format of "<UTTERANCE> | <CONFIDENCE> | <SPEAKER> | <RATIONALE> | <IMAGE DESCRIPTION>".
(2) you MUST select the utterance in the given dialogue, NOT generate a new utterance.
(3) the rationale should be written starting with "To".
(4) your answer should be written starting with "An image of" and in one sentence.
(5) you do NOT include the speaker's name (i.e., `[speaker1]`, `[speaker2]`) in the image description.
(6) you should share a realistic image, NOT memes.

Answer:
1.

Figure 13: **Prompt Templates for One Stage.** A prompt template for one stage framework.

**Prompt Template for Stage 1:**
The following is a dialogue between `[speaker1]` and `[speaker2]`. You should share an image to make the following dialogue more interesting and engaging. The dialogue is provided line-by-line. In the given dialogue, select all utterances that are appropriate for sharing the image in the next turn, and write the speaker who will share the image after the selected utterance. You should also provide a rationale for your decision.

Dialogue:
`[dialogue]`

Restrictions:
(1) your answer should be in the format of "<UTTERANCE> | <SPEAKER> | <RATIONALE>".
(2) you MUST select the utterance in the given dialogue, NOT generate a new utterance.
(3) the rationale should be written starting with "To".

Answer:
1.

Figure 14: **Prompt Templates for Stage 1 (w/o Confidence-based Ranking).** A prompt template used in the ablation study in Stage 1 for validating the effect of confidence-based ranking.

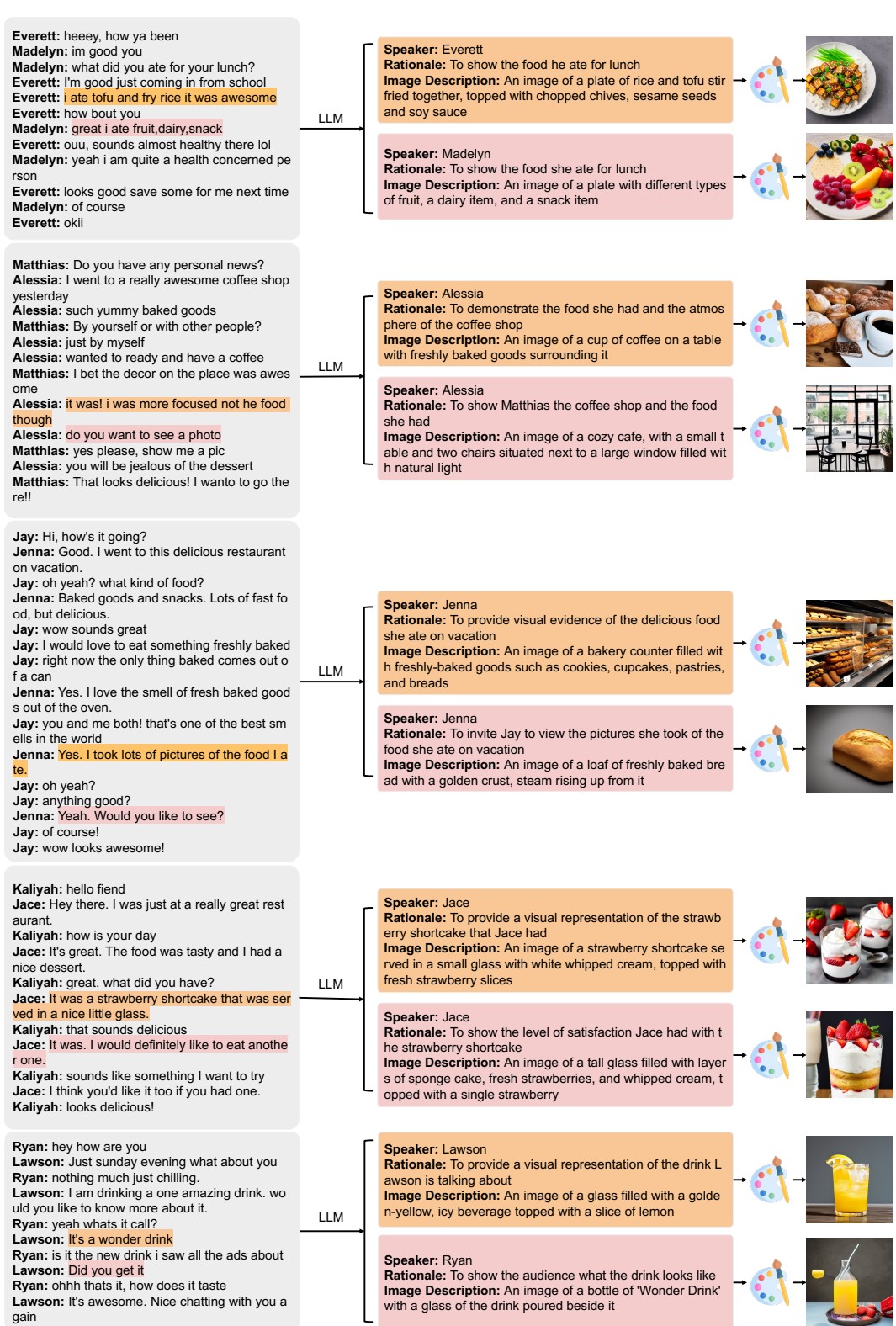

Figure 15: **More Examples of PhotoChat++.** We present more generated examples of PhotoChat++ dataset using our proposed framework with LLM (i.e., text-davinci-003) and Stable Diffusion(🎨).

## L  REGEX PATTERN

For stage 1, we use the regex pattern to extract the `utterance`, `confidence`, `speaker`, and `rationale` as follows:

```
^(?:\d+\.\s+)?\"?(?P<utterance>.*?)\"?\s+\|\s+(?P<confidence>.*?)\s+\|
\s+(?P<speaker>.*?)(?:\s+\|\s+(?:Rationale:\s+)?(?P<rationale>.*?))?$
```

