# OpenReview forum: "Large Language Models can $\textit{Share}$ Images, Too!"
_ICLR.cc/2024/Conference — ICLR 2024 Conference Withdrawn Submission_

### Official Review · Reviewer_7gME · 2023-10-15

**Soundness:** 1 poor
**Presentation:** 3 good
**Contribution:** 1 poor
**Rating:** 3
**Confidence:** 4

**Summary:**

This submission is primarily concerned with the “image sharing” capability of LLM, to make it a more human-like chatbot. It proposes a two-stage mechanism for image sharing which is based on prompt-engineering.

**Strengths:**

The task is novel and critical for a chatbot application.

**Weaknesses:**

The task needs to be better motivated:
1.	Though “sharing” images is very well-motivated for today’s chatbot applications. If it’s a retrieval task over Internet images, it could bring user with complimentary information missing from language. However, if it’s as shown in one of the expriments, where image generation conditioned on texts which are generated from LLM is shared, I’m afraid there were nothing language could not deliver in the first place.
2.	Even though there is a previous work, Zang et al., who proposes the PhotoChat dataset, it’s still worth questioning if the “two-stage” framework is correct, or a reasonable approximation of how we should approach the capability. I found it unconvincing to me and is pending more justification.

The method is plain and lacks comparison with alternatives:
3.	I doubt any readers who have played with modern LLM would be surprised/informed by this work that LLM can already predict whether to share images and provide image descriptions accordingly.
4.	What about some other finetuning-based methods?


Following this, there are two critical scientific errors in this submission:
5.	It’s confusing to say “we present a restriction-based prompt by adding a Restrictions: token.” This reads to me as if you introduced a new token to the vocabulary in a finetuning stage and might confuse others. In fact, it’s just typing in the word “Restrictions: ” According to [this link](https://platform.openai.com/tokenizer), this is even split into 4 tokens.
6.	Claiming image-sharing an emergent capability isn’t quite right, considering the authors are not observing the model sizes in a sufficiently fine-grained fashion as in Wei et al. Also the Figure 2 should be changed into a table. It’s scientifically sloppy to plot the x-axis as it is now. Only comparable values should appear in x-axis of a line chart.

**Questions:**

N/A

---

### Official Review · Reviewer_x6ME · 2023-11-02

**Soundness:** 1 poor
**Presentation:** 3 good
**Contribution:** 2 fair
**Rating:** 3
**Confidence:** 4

**Summary:**

In this paper, the authors study the ability of pure LLM-based dialogue system to understand when and where suitable images should be shared to enrich a conversation.
This is tackled through a two-stage system, which leverages particularly structured input prompt templates. In the first stage, the prompt template is designed to encourage the LLM to produce a ranking of dialogue utterances based on a LLM-designated confidence score. In a second stage, the authors aim for suitable positions to be replaced by detailed image descriptions, which later on may be converted to realistic-looking images via text-to-image generative models.
For their proposed prompt-template-based setup, various experiments are conducted, mainly aiming to compare different proprietary and open LLMs. On this basis, the authors draw the conclusions that image-sharing capacities may be of size-emerging nature, and that LLMs on their own can suitable judge where and what images to incorporate in a dialogue.

**Strengths:**

* I believe that the overall goal of separating the process of image generation and its definition and placement within a dialogue has merits, as it introduces a degree of modularity and interpretability (by operating entirely in the human-understandable text domain), which allows image generation to be done with any text-to-image model of choice.
* To the best of my knowledge, this is also the first work that aims to tackle multi-modal dialogue from such an entirely modular perspective.

**Weaknesses:**

I have several issues and questions with this work, which I have separated into those I find most important to have addressed, and those that are good to tackle to raise the overall quality of the work.

__Larger issues__

* Fundamentally, vision foundation models or fundamentally multimodal models are still required to generate the final images for a multi-modal dialogue. As such, the restriction to LLM-only strategies should be appropriately supported, as it otherwise appears like a fairly contrived research problem. Unfortunately, I do not find any comparison to existing multi-modal dialogue models or even a discussion on their limits (as listed e.g. Zhang et al. 2021 or Koh et al. 2023). It would be great if the authors could address this.

* It's also difficult to determine the significance of the contribution.

	* For example, the authors introduce a new prompt template to tackle this problem. But limited details are missing, with no examples for full prompts provided, or additional details such as the potential use of in-context examples. Particularly without the latter, I find it hard to believe that the LLM naturally produces a suitable ranking of confidence-based candidate positions. Similarly, how can the authors guarantee that the produced confidence has any actual meaning?

	* A lot of claims are made without sufficient experimental backing. For example:

		* p.5: "... a scaling law (...) also exists in the image-sharing turn prediction task.". I'm not sure if the data-point and their size-relation allow for suitable conclusions over scaling laws, as the parameter sizes go from 6.7B by nearly two magnitudes to 175B, and then also includes same-sized variants trained differently.

		* For the human evaluation, what are the exact metrics used to measure turn relevance/speaker adequacy? What are the exact setups for the human evaluation studies, e.g. the number of participants, the experimental setup, what is the significance of the results, etc.? This impacts both Fig. 5 and Fig. 6, where results are in parts very close to each other (e.g. w/rank & w/o rank).

		* "LLMs generate roughly 3.1 image-sharing turns (...), a reasonable count compared to 2.59 turns in multi-modal dialogue datasets" - this is a difficult statement to make when looking at table 3, which shows large variances in the number counts (between 1.58 and 6.8).

		* The motivation and relevance of the diversity comparisons w.r.t. to the PhotoChat baseline: Is it not the case that PhotoChat only has one image per dialogue, while the LLM systems are allowed to generate multiple ones? This would throw off the direct comparability.

		* For 2.3, the authors simply mention a crucial initial prompt template ablation study, but provide no further details. However, numerical comparisons are important to understand the significance of the particularly proposed prompt template.


__Smaller issues__

* Connecting points in Fig. 2 implies a sequential relation between the tested models, which does not exist.

* Why the consistent change between F1@All, F1@K and F1@1 across different plots (e.g. Fig.2 -> Fig.3 (right))?

* Fig. 4 doesn't seem like the most suitable example, as the dialogue could be interpreted as concluded as well.

* CLIPScore is somewhat lacking for meaningful text-to-image alignment, methods such as ImageReward may be more suitable.

* Section 2.2 is redundant, and should be absorbed into 2.1, e.g. as a paragraph.

* It would be great to provide an example of how the prompt template is filled in practice in the main paper.

* The emoji in the 2.3 title (and generally when referencing their restriction-based template) is somewhat confusing, as it appears to imply "no restrictions".

* Page 3: The change in indices for s_t -> (s_j)^(t-1)_1 makes it slightly harder to parse

**Questions:**

I am currently opting for rejection, as I don't believe the problem setting to be sufficiently motivated, and the experiments to provide sufficient evidence (or even a lack thereof). I am however willing to raise my score if the authors can address the larger issues mentioned in the previous section, particularly with respect to the importance and relevance of this problem (as vision-foundation models / multi-modal ones will still have to be queried inevitably), as well as the experimental significance.

---

### Official Review · Reviewer_LPEG · 2023-11-05

**Soundness:** 2 fair
**Presentation:** 3 good
**Contribution:** 2 fair
**Rating:** 5
**Confidence:** 2

**Summary:**

This paper studies the image-sharing capability when using Large Language Models (LLMs), such as InstructGPT, ChatGPT, and GPT-4, in a zero-shot setting, without the help of visual foundation models. Inspired by the two-stage process of image-sharing in human dialogues, the paper proposes a two-stage framework that allows LLMs to predict potential image-sharing turns and generate related image descriptions using the proposed effective restriction-based prompt template. The paper uncovers the emergent image-sharing ability in zero-shot prompting, demonstrating the effectiveness of restriction-based prompts in both stages of the proposed framework.

The paper also creates a new dataset, and the paper plans to share the dataset together with the code after publication.

**Strengths:**

(1) The studied problem/use case is very interesting.

(2) The paper is well written and it is very easy to read/follow.

(3) The proposed method has good intuitions.

**Weaknesses:**

(1) It is not very clear why the studied setting should assume 'without the help of visual foundation models.', considering (a) it is so easy to access the visual foundation models these days (and the large multimodal model seems to be a potential future trend, e.g., GPT-4) and (b) many similar previous works (mentioned in the 2nd paragraph of the introduction section) use the visual foundation models.

(2) A potential related concern is that all the baselines (except PhotoChat Zang et al. (2021)) in this paper seem to not be from previous works? Are there any baselines from previous papers can be compared here? Is it possible to compare the results in the paper to the results by the previous works mentioned in the 2nd paragraph of the introduction section, in some aligned setting?

**Questions:**

(1) Can you please justify why it is needed to assume 'without the help of visual foundation models.' (e.g., by some practical needs or real-world use cases)

(2) Is it possible to compare the results in the paper to the results by the previous works mentioned in the 2nd paragraph of the introduction section, in some aligned setting?

---

### Official Review · Reviewer_aJto · 2023-11-06

**Soundness:** 3 good
**Presentation:** 2 fair
**Contribution:** 2 fair
**Rating:** 5
**Confidence:** 4

**Summary:**

The paper explores the image-sharing capability of Large Language Models (LLMs) in a zero-shot setting, without the help of visual foundation models. The problem involves two stages: (1) when to share and (2) what to share. The evaluate the method, the authors augmented the PhotoChat dataset.

**Strengths:**

The proposed method is technically sound.

Doing visual understanding without any help of visual foundation models is interesting.

The proposed method is effective.

**Weaknesses:**

The proposed method seems a bit straightforward and not that novel.

Although the problem to understand vision without visual models is interesting, the image-sharing application itself is not that impressive.

**Questions:**

Besides dialog, can the image-sharing capability be applied to other scenarios?

---

### Official Review · Reviewer_m6aj · 2023-11-07

**Soundness:** 2 fair
**Presentation:** 2 fair
**Contribution:** 3 good
**Rating:** 3
**Confidence:** 4

**Summary:**

This paper focuses on the image-sharing capability of Large Language Models without the help of visual information.  To this end, they propose a two-stage framework: (1) predicting image-sharing turns and (2) generating image prompts, to mimic the human photo-sharing behavior. Specifically, they design a restriction-based prompt for unlocking the ability of image-sharing. Therefore, they conduct extensive experiments on PhotoChat++ (augmented by the PhotoChat), which show that LLMs achieve competitive zero-shot performance without additional training.

**Strengths:**

1.  **Clear Motivation**. The motivation for unlocking the image-sharing capability in the frozen LLM is clear and interesting.

2. **Contribution of the new dataset**. The photochat++ dataset may be meaningful for image-share ability if this dataset can be released.

**Weaknesses:**

1. **The writing is bad.** The abstract section is so confusing to me that it has many undefined nouns, such as *What is image-sharing capability?* "Inspired by the two-stage process of image sharing, what is the two-stage process?" Figure 1 makes it hard to understand the key point of image-sharing ability.

2. **The technical contribution is limited**. In the method section, it seems that you just use the open API and a prompt engineer to predict when to share and what to share. It can not provide more technical insight for the community.

3. **The experiment is not enough**. Apart from LLM API, the open-source LLM models (such as Vicuna, llama v1, llama v2 [1]) also need to be evaluated.  It is important to compare the existing methods in photochat benchmark.

4. **The necessity of the image-shared ability in the LLM**. With the growth of the Large Multimodal Models (LMM) (e.g. MiniGPT4[2], LLava[3], Openflamingo[4], Emu[5]), if it is necessary to unlock the image-shared ability in LLM, rather than directly use LMM to predict when to share and how to share.

[1] https://github.com/lm-sys/FastChat

[2] Zhu, Deyao, et al. "Minigpt-4: Enhancing vision-language understanding with advanced large language models."

[3] Liu, Haotian, et al. "Visual instruction tuning."

[4] Awadalla, Anas, et al. "Openflamingo: An open-source framework for training large autoregressive vision-language models."

[5] Sun Q, Yu Q, Cui Y, et al. Generative pretraining in multimodality.

**Questions:**

As shown in weaknesses.